# Wheat Bran and Saccharomyces Cerevisiae Biomass’ Effect on Aerobic and Anaerobic Degradation Efficiency of Paper Composite

**DOI:** 10.3390/microorganisms12102018

**Published:** 2024-10-05

**Authors:** Zita Markevičiūtė, Arianna Guerreschi, Glauco Menin, Francesca Malpei, Visvaldas Varžinskas

**Affiliations:** 1Centre for Packaging Innovations and Research, Kaunas University of Technology, 51424 Kaunas, Lithuania; 2Fabe Laboratory, Department of Civil, Environmental and Infrastructure Engineering, Politecnico di Milano, 20156 Milan, Italy; 3Environmental Engineering Laboratory, Department of Civil, Environmental and Infrastructure Engineering, Politecnico di Milano, 20156 Milan, Italy; 4Department of Civil, Environmental and Infrastructure Engineering, Politecnico di Milano, 20156 Milan, Italy; 5Institute of Environmental Engineering, Kaunas University of Technology, 44239 Kaunas, Lithuania

**Keywords:** *Saccharomyces cerevisiae* application in paper packaging, biodegradable packaging material, industrial by-products applications, wheat bran application in paper packaging

## Abstract

This study is a continuation of research on sustainable food packaging materials made from locally available feedstock and industrial by-products within the Baltic Sea region. Its main focus is the impact of wheat bran filler and *Saccharomyces cerevisiae* additive, which was used to develop a novel bio-coating for paper composite packaging, on the biodegradation efficiency of paper composites under aerobic and anaerobic conditions. In this study, we analyzed the effect of 15% and 40% concentrations of wheat bran filler and *Saccharomyces cerevisiae* biomass on the biodegradation efficiency of paper composites. This research was conducted under controlled environmental conditions, with aerobic biodegradation tested at 46 °C in a compost-based mesophilic–thermophilic environment and anaerobic biodegradation tested at 55 °C in an active inoculum thermophilic environment. The results show that the presence of wheat bran filler significantly improves biodegradation efficiency compared to microcrystalline cellulose reference material. Under aerobic conditions, the biodegradation efficiency for the 40% wheat bran and yeast sample was 6.34%, compared to only 0.71% for the cellulose reference material. In anaerobic conditions, the 15% wheat bran and yeast sample showed a biodegradation efficiency of 96.62%, compared to 82.32% for the cellulose reference material.

## 1. Introduction

This study builds upon prior research aimed at developing fully green food packaging material sourced from locally available feedstock and industrial by-products within the Baltic Sea region. Previous stages of research have identified the benefits of bio-based and biodegradable fast food packaging for the circular economy [1], analyzed locally available sustainable feedstock materials [2], and tested material samples for physical–mechanical- and barrier-related properties [3]. The findings from previous studies highlight several key points, as follows:(a)Biodegradable food packaging offers significant benefits to the circular economy by purifying waste streams, diverting food waste from landfills, and providing sustainable alternatives for non-recyclable packaging [4]. This approach facilitates the return of biomaterials to the soil, thereby enhancing biological recycling benefits [5,6,7,8].(b)The utilization of plant-based feedstocks, particularly those sourced from industrial waste or by-products, highlights the importance of renewable resources with lower environmental impacts [2].(c)The application of paper (both wood and non-wood fiber) in fully green food packaging remains limited due to stringent barrier property requirements. The natural components of yeast, such as proteins and glucans, have unique characteristics that could enhance fully green food packaging barrier properties.

Several studies have shown that *Saccharomyces cerevisiae* biomass can benefit uncoated paper food packaging due to its potential for biofilm formation [9,10,11,12]. Additionally, it can benefit coated paper packaging by impacting plastic degradation [13,14] and microplastic removal [15]. Seeking to improve the barrier properties of the paper composite created in a previous experimental stage of research [3], *Saccharomyces cerevisiae* was added as a potential biofilm-forming additive. The observed results indicated that the addition of yeast affected barrier properties variably: it increased air permeability while enhancing surface hydrophobicity. The material samples that demonstrated the best hydrophobicity properties were selected for further research.

This study aims to investigate the impact of wheat bran filler and added *Saccharomyces cerevisiae* on material biodegradation efficiency. Key factors influencing material biodegradation may include the specific composition of biodegradable packaging materials (concentration of wheat bran filler and *Saccharomyces cerevisiae* biomass), the environmental conditions under which biodegradation is tested, and the interaction between the material and microbial communities in the biodegradation environment. Maintaining consistent environmental conditions is essential for both aerobic and anaerobic digestion to achieve their full potential. In both environments, optimizing key factors such as temperature, pH levels, substrate composition, and nutrient availability is critical for enhancing microbial activity [16,17,18,19,20,21,22,23,24,25,26]. Therefore, this study covers the analysis of the impact of wheat bran filler concentration and *Saccharomyces cerevisiae* biomass on material biodegradation under different temperatures and inoculum conditions: aerobic biodegradation at 46 °C in a compost-based environment, and anaerobic biodegradation at 55 °C in an active inoculum environment.

The key impact of this research is the identification of optimal conditions and wheat bran filler compositions for enhancing the biodegradation efficiency of paper composites, as well as providing insights into the biodegradation potential of yeast biomass. The findings demonstrate that the inclusion of *Saccharomyces cerevisiae* and higher concentrations of wheat bran significantly improve the biodegradation process, offering valuable insights for the development of sustainable and efficient biodegradable packaging materials. However, the validity of the aerobic test is compromised due to the insufficient degradation of the reference material (0.71% in 30 days), indicating the need for further testing under improved conditions to ensure accurate and reliable results.

## 2. Materials and Methods

### 2.1. Test Samples

Paper composite samples (see Table 1) that demonstrated the best hydrophobic properties in a previous research stage [3] were selected for aerobic and anaerobic biodegradation efficiency testing. Different filler compositions were chosen to compare the effect of bran on material biodegradation efficiency. For analyzing the effect of yeast, samples containing the same concentration of bran, with and without yeast, were selected. The detailed materials description, chemical composition, and production procedure of the paper composite samples are provided in our previous publication [3].

### 2.2. Inoculum

Compost used for the aerobic test was taken from a composting plant north of Milan treating selected biowaste and green waste. The anaerobic inoculum was taken from a thermophilic digestor operating at 50 °C in Carimate WWTP (Como, Italy).

### 2.3. Aerobic Biodegradation Tests

Tests were performed in the LIA lab of DICA Politecnico di Milano. Laboratory equipment used in the study included the OxiTop^®^ respirometric system (WTW, Xylem Analytics, Weilheim, Germany) for measuring biological oxygen demand (BOD) under aerobic conditions. Well-aerated compost from a properly operating aerobic composting plant was prepared and used as inoculum following ISO 14855-2 [27].

The inoculum was further diluted with an inorganic test medium prepared following ISO 14851:2019 [28]. Three types of dry test substrates were cut into 10 × 10 mm squares and then added to the diluted inoculum. Microcrystalline cellulose was used as a reference material. Detailed substrate and inoculum characterization are provided in Table 2.

For testing, 1110 mL volume glass vessels were used, and sodium hydroxide was present in the top cap for CO_2_ absorption. To ensure even mixing and temperature homogeneity during BOD testing, vessels were placed on the IS 12 inductive stirring system. Test vessels were incubated in a temperature-controlled oven, with gentle continuous mixing using a magnetic stirrer. The temperature was set at 46 °C throughout the entire 30-day incubation period. To prevent oxygen deficiency, air was supplied to the test vessels via an aerator for 5 min every 5–6 days.

The selected temperature is in the range of optimal composting temperatures [29], where both mesophilic and thermophilic bacteria are active [30], and is optimal for microbial activity and pathogen elimination [31,32].

Substrate, blank assays, and positive controls were carried out in triplicates.

The aerobic biodegradability achieved for each test specimen at the end of the test was calculated as follows—the oxygen pressure values generated from the substrate (hPa/bott) were obtained from the respirometric measuring system controller data. Every 14 days, 360 measurements were generated, resulting in a total of 772 measurements during the entire incubation period. The data were transferred to a PC and analyzed afterward. The measured pressure obtained with respirometric methods was converted into the BOD value using the following equation:BOD = (M(O_2_)/(R * T_*m*_)) * ((V_*tot*_ − V_*l*_)/V_*I*_) * α * (T_*m*_/T_0_) * Δ_*p*_(O_2_)
$$BOD=\frac{M\left(O_{2}\right)}{RT_{m}}\left(\frac{V_{tot}-V_{1}}{V_{1}}\right)\alpha \left(\frac{T_{m}}{T_{0}}\right)\Delta p(O_{2})$$
where

M(O_2_) = Molecular weight of oxygen (32,000 mg/mol)R = Gas constant (83.144 L·hPa/(mol·K))T_0_ = Temperature (273.15 K)Tm = Measuring temperature (319.15 K for performed BOD)Vtot = Bottle volume [mL]V1 = Sample volume [mL]α = Bunsen absorption coefficient (0.03103)Δp(O_2_) = Difference in the partial oxygen pressure [hPa]

The test results represent the maximum level of biodegradation determined from the plateau stage of the biodegradation curve [28].

The specific biological oxygen demand of each tested material (BOD_total_) was calculated as the difference between oxygen consumption in the test flasks and the blanks.

Given that the COD of the sample added to each bottle (COD_sample_) represents the total oxygen demand for the chemical oxidation of the organic matter inserted, the oxygen demand obtained in long-duration aerobic biodegradation tests like the ones here performed (BOD_total_), is a close proxy of the biodegradable organic substrate in the bottle (COD_sample bio_) and the ratio of BOD_total_/COD_samples_ gives a close proxy of the fraction COD_sample bio_/COD_sample_ and of the biodegradability of the organic compound present in the bottle.

A more accurate calculation of the CODbio according to the COD mass balance principles, however, should have considered that a part of the COD_sample bio_ is converted to bacterial cells by growth (without oxygen consumption), and a fraction of this remains at the end of the test as unbiodegradable cell residuals from bacterial decay. Finally, BODtotal values always slightly underestimate the quantity of biodegradable organics in the sample (CODbio). Formula to calculate CODbio from BODtotal are well described in the literature [33] and have been here applied, assuming that 10% of the initial CODbio remains as unbiodegradable organics residues, as follows:$$Biodegradation\;efficiency\left(\%\right)=\left(0.9*{BOD}_{total}/{COD}_{sample}\right)*100\;=\left(\frac{0.9\times {BOD}_{total}}{{COD}_{sample}}\right)\times 100$$

### 2.4. Anaerobic Biodegradation

Tests were performed in the Fab-e lab of DICA Politecnico di Milano. Laboratory equipment used in the study included Gas Endeavour III (GE III, BPC Instruments, Lund, Sweden) for measuring methane production under anaerobic conditions.

The anaerobic inoculum was left without feeding for 7 days (degassing). Afterward, according to Italian standard UNI/TS 11703:2018 [34], nutrient media were added, and the inoculum was further diluted. Italian standard UNI/TS11703:2018 was used along with the best BMA test practices [35].

Three types of dry test substrates, cut into 10 × 10 mm squares, were then added to the diluted inoculum. Microcrystalline cellulose was used as a reference material. The final concentration ratio of total solids in the vessel and the pH at the start of the test were adjusted to 7, as described in ISO 14853 [36]. The mass of the substrate to be dosed was calculated by dividing the mass of the substrate to be dosed vs. by the volatile solids to total solids ratio (VS/TS-S) and was precisely scaled. Detailed substrate and inoculum characterization, allowing for a maximum error of 1%, is provided in Table 3.

Biomethane potential (BMP) tests were carried out using a Gas Endeavour III (GE III, BPC Instrument, Sweden) consisting of 18 units of 500 mL volume reactors and the same number of gas flow meters (flow cells) attached to a detection unit for automatic data acquisition. BMP tests were conducted under thermophilic conditions at 55 °C [37,38,39]. Test vessels were incubated in a temperature-controlled water bath, with gentle and continuous (1 min running, 2 min stop) mixing applied to the vessels throughout the entire 30-day incubation period [40,41].

Substrate, blank assays, and positive controls were carried out in triplicates.

The anaerobic biodegradability achieved for each test specimen was calculated as follows—from the total cumulative methane values generated from each test (NmLCH_4_/bott), the net methane values were calculated by deducting the blank vessel value. Then, the net values were normalized to the COD of the substrate added (NmLCH_4_/mgCOD) (CH_4_ final). Then, the biodegradability achieved during the test was calculated by dividing the normalized production by the equivalence factor α: 0.32 mL CH_4produced_/mg COD_biodegraded_ considering 10% of the substrate COD is consumed for biomass growth and is not giving any methane production [42].

Biodegradation efficiency is then calculated based on the amount of methane produced compared to the substrate COD added at the beginning of the test according to the following formula:Biodegradation efficiency (%) = (CH_4*Final*_/Theoretical CH_4_) * 100
$$=\left(\frac{\;{CH}_{4}\;Final}{{Theoretical\;\;CH}_{4}}\right)\times 100$$

### 2.5. Analytical Determination

Total solids (TS) and volatile solids (VS) were determined following ISO 14855-2 [27].

Chemical Oxygen Demand (COD) was determined using the Dichromate Method [43]. The COD values obtained were used as the initial COD in the anaerobic material biodegradation data analysis.

## 3. Results and Discussion

### 3.1. Aerobic Biodegradation Results

The data in Table 4 present the summary of measurements related to the aerobic biodegradation process of different samples involving bran filler and yeast additive. The key metrics include the initial theoretical Chemical Oxygen Demand (COD) determined using the Dichromate Method (see Section 2.5), the COD (mg O_2_/bottle) generated during the incubation period, and the final COD with deducted new microorganism growth and repair activity, and calculated biodegradation efficiency, allowing for a maximum error of 1%.

The reference sample shows minimal change, indicating a low biodegradation of cellulose (0.71%).

Wheat bran concentration appears to have a notable impact on biodegradation. Wheat bran is rich in non-starch polysaccharides, such as arabinoxylans and β-glucans, which provide a good substrate for microbial activity. These components enhance the growth and activity of microorganisms involved in biodegradation, leading to increased breakdown of organic material. Additionally, wheat bran contains various essential nutrients that support microbial metabolism, making it a valuable additive in biodegradation processes [44].

The B15Y sample shows a modest increase in BOD, suggesting a lower biodegradation rate compared to samples with higher wheat bran concentrations. The biodegradation efficiency for this sample is 2.24%. The sample B40Y demonstrates a higher BOD increase compared to B15Y, indicating a higher biodegradation rate. The biodegradation efficiency for this sample is 4.07%. Increasing the amount of wheat bran in the biodegradation process provides more substrate for microorganisms, which can lead to more efficient degradation of organic matter. Higher concentrations of wheat bran also mean more available nutrients and structural polysaccharides [45,46], which can stimulate microbial activity and enhance the overall biodegradation rate.

Yeast addition seems to enhance biodegradation. The 40% wheat bran sample without yeast shows a biodegradation efficiency of 4.07%, while the 40% wheat bran sample with yeast exhibits the highest BOD increase among the samples, suggesting that yeast enhances biodegradation. The biodegradation efficiency for this sample is 6.34%. *Saccharomyces cerevisiae* biomass contains a variety of nutrients such as proteins, vitamins, and minerals, which can be released into the environment and used by other microorganisms. These nutrients can enhance the growth and activity of bacteria and other microbes that are active in the biodegradation process, leading to improved efficiency [47]. Yeast cell biomass can also provide structural support and serve as a physical matrix for microbial colonization [48]. The physical presence of yeast cells can thus contribute to a more effective breakdown of substrates by supporting microbial communities.

In general, the data indicate that higher concentrations of wheat bran result in higher biodegradation efficiencies. Additionally, the presence of yeast enhances the biodegradation process, as evidenced by the highest efficiency observed in the 40% wheat bran with yeast sample. This sample demonstrated a 35% higher biodegradation rate compared to the sample without yeast.

Aerobic biodegradation test validity: the validity of the aerobic test is questionable due to the exceptionally low biodegradation efficiency of the reference material (cellulose), which degraded by only 0.71% over 30 days. According to ISO 14855-2 [27], the test is considered valid only if the degree of biodegradation of the reference material exceeds 70% after 45 days. Given that the biodegradation rate falls significantly below this threshold, the low biodegradation rates suggest that the test environment, inoculum, or equipment [49] may not have been optimal for supporting aerobic microbial activity. Consequently, the aerobic biodegradation efficiency of other materials tested in this environment may also be underestimated.

### 3.2. Anaerobic Biodegradation Results

The data in Table 5 present the summary of measurements related to the BMP anaerobic biodegradation process of different samples involving bran filler and yeast additive. The key metrics include Chemical Oxygen Demand (COD), theoretical and final methane (CH_4_) production, and biodegradation efficiency.

Figure 1 and Table 6 present performed materials anaerobic biodegradation test mean net cumulative gas production (NmL CH_4_) over time.

All material sample treatments show a rapid increase in gas production within the first 5 days, followed by a plateau phase.

The sample with 15% bran filler and yeast additive shows the highest gas production, reaching around 883.60 NmL CH_4_, maintaining the highest level throughout the period. The sample with 40% bran filler and yeast additive shows slightly lower gas production, indicating that increasing the bran content reduces material biodegradation. The observation that lower bran filler content (15%) with yeast additive is more efficient for biodegradation and methane production compared to higher bran filler content (40%) can be influenced by different lignin content and other factors. Lignin is known to be resistant to microbial degradation, particularly in the absence of oxygen, due to its complex structure [50,51,52,53]. Therefore, higher lignin content can inhibit the biodegradation process [54], and lignin concentration in the tested paper composites correlates negatively with biodegradation efficiency [55].

Another factor might be microbial synergy. The interaction between different microbial communities (bacteria, archaea, and fungi) is crucial for efficient biodegradation [56,57,58]. A balanced substrate composition (like 15% bran filler) may support a more synergistic microbial environment compared to a higher bran content (40%).

Even though increasing bran filler content from 15% to 40% slightly reduces the gas production (when yeast additive is present), both bran filler contents show significantly higher gas production compared to the cellulose reference, indicating that cellulose alone is less effective for methane production compared to bran-based composites.

The B40 sample with 40% bran filler and no yeast additive shows slightly lower (826.30 NmL CH_4_) gas production compared to B40Y, which is the same bran content with yeast additive (847.40 NmL CH_4_), indicating a non-significant positive effect of the yeast additive on gas production. On the other hand, B40Y biodegradation is 88.14%, slightly lower compared to B40’s 88.50%. This could be explained by initial rapid degradation in the beginning and incomplete degradation. The presence of yeast additive could enhance the initial microbial activity, rapidly breaking down easily accessible organic matter and producing a significant amount of gas early in the process. Once the easily degradable components are consumed, the remaining substrate may not be as readily accessible or degradable by the microbial community, resulting in lower overall biodegradation.

## 4. Conclusions

*The comparison of aerobic and anaerobic degradation:* The performed tests reveal distinct differences in efficiency and outcomes. Under aerobic conditions, the biodegradation efficiency is significantly lower compared to anaerobic biodegradation, with values ranging from 0.71% to 6.34%. B40Y exhibits the highest aerobic efficiency at 6.34%, followed by B40 (4.07%) and B15Y (2.24%). Cellulose shows the lowest efficiency at 0.71%.

In contrast, anaerobic biodegradation shows much higher efficiencies, with biodegradation percentages reaching up to 96.62% for B15Y, while B40 and B40Y demonstrate slightly lower values of 88.50% and 88.14%, respectively. However, these values are still significantly higher compared to the control (cellulose, 82.32%).

This stark difference can be attributed to the metabolic capabilities of anaerobic microbes, which efficiently convert organic matter into biogas, particularly methane (CH_4_).

*Aerobic biodegradation (46 °C, Mesophiles and Thermophiles):* Higher concentrations of wheat bran lead to higher biodegradation efficiencies due to the rich non-starch polysaccharides and essential nutrients present in wheat bran, which support and stimulate both mesophilic and thermophilic microbial activity.

*Anaerobic biodegradation (55 °C, Thermophiles)*: Thermophiles at 55 °C have fewer metabolic pathways available for degrading organic matter without oxygen. The breakdown of organic matter is slower, and the presence of lignin in wheat bran further complicates the process due to its resistance to microbial degradation.

*The addition of yeast:* Even in an inactive form yeast additive provides additional nutrients that might stimulate microbial growth and activity, enhancing biodegradation efficiency under aerobic conditions and having an insignificant positive impact in anaerobic conditions.

*Future research:* Due to the observed exceptionally low aerobic biodegradation efficiency, future research will focus on optimizing test conditions, and testing materials at a temperature range of 50–60 °C, using compost with a 40–60% moisture content as the inoculum. Advanced testing equipment with an automatic oxygen supply will be utilized.

## Figures and Tables

**Figure 1 microorganisms-12-02018-f001:**
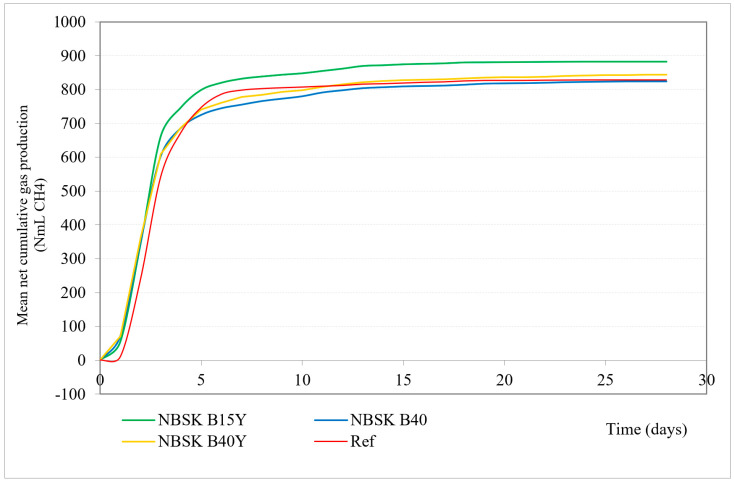
Mean net cumulative gas production (NmL CH_4_) in 30 days. Green line: 15% bran filler, yeast additive; yellow line: 40% bran filler, yeast additive; red line: reference/cellulose; blue line: 40% bran filler, no yeast additive.

**Table 1 microorganisms-12-02018-t001:** Materials selected for aerobic and anaerobic tests.

Substrate ID	Bran Content (%)	*Saccharomyces cerevisiae*
B15Y	15	Yes
B40	40	No
B40Y	40	Yes

**Table 2 microorganisms-12-02018-t002:** Materials characterization and mean values of aerobic degradation test.

	VS (Per Unit Mass)	VS/TS	Mass, Material	Mass, VS	COD	Volume of Test Medium	Working Volume
**Substrate ID**	(gVS, S/kgFM, S)	(%)	(gFM, S/bott)	(gVS, S/bott)	mgO_2_/bott	(mL)	(mL)
B15Y	940.77	99.71	1.0025	0.94	1065.62	200	217
B40	927.09	98.92	1.0049	0.93	1076.95	200	217
B40Y	932.36	99.62	1.0019	0.93	1110.41	200	217
Cellulose	960.71	99.99	1.0027	0.96	1188.20	200	217
**Inoculum**							
Compost	182.27	38.15	20.00	3.65	-	-	-

**Table 3 microorganisms-12-02018-t003:** Materials characterization and mean values of anaerobic degradation test.

	VS (Per Unit Mass)	VS/TS	Mass, Material	Mass, VS	COD	Volume of Dilution Tap Water	Total Volume of Nutrients	Working Volume
**Substrate ID**	(gVS, S/kgFM, S)	(%)	(gFM, S/bott)	(gVS, S/bott)	mgO_2_/bott	(mL)	(mL)	(mL)
B15Y	940.77	99.71	2.42	2.28	2572.36	74.80	52.8	480.00
B40	927.09	98.92	2.45	2.27	2625.67	74.80	52.8	480.00
B40Y	932.36	99.62	2.44	2.27	2704.25	74.80	52.8	480.00
Cellulose	960.71	99.99	2.39	2.30	2832.15	74.80	52.8	480.00
**Inoculum**								
Sludge	13.77	57.35	350.00	4.82	-	-	-	-

**Table 4 microorganisms-12-02018-t004:** The aerobic test data.

Substrate ID	Initial COD (mgO_2_/Bott)	COD from Test (mgO_2_/Bott)	COD Corrected (mgO_2_/Bott)	Initial COD (mgO_2_/L)	COD from Test (mgO_2_/L)	Coefficient of Variation CV(%)	COD Corrected (mgO_2_/L)	Biodegradation Efficiency (%)
B15Y	1065.62	26.56	23.91	4910.69	122.40	7.11	110.18	2.24
B40	1076.95	48.64	43.78	4962.90	224.15	5.66	201.75	4.07
B40Y	1110.41	78.26	70.43	5117.10	360.65	3.83	324.56	6.34
Cellulose	1188.20	9.33	8.40	5475.58	43.00	17.25	38.71	0.71

**Table 5 microorganisms-12-02018-t005:** The anaerobic test data.

Substrate ID	Initial COD (mgO_2_/Bott)	NET CH_4_ Produced (mL/Bott)	CH_4_ Final (mL/Bott)	Theoretical CH_4_ (mL/Bott)	Initial COD (mgO_2_/L	NET CH_4_ Produced (mL/L)	Coefficient of Variation CV(%)	CH_4_ Final (mL/L)	Theoretical CH_4_ (mL/L)	Biodegradation Efficiency (%)
B15Y	2572	883.6	795.24	823.04	5358.33	1840.83	2.34	1656.75	1714.67	96.62
B40	2626	826.30	743.67	840.32	5470.83	1721.46	1.41	1549.31	1750.67	88.50
B40Y	2704	847.40	762.66	865.28	5633.33	1765.42	1.7	1588.88	1802.67	88.14
Cellulose	2832	828.90	746.01	906.24	5900.00	1726.88	16.7	1554.19	1888.00	82.32

**Table 6 microorganisms-12-02018-t006:** Mean net cumulative gas production (NmL CH_4_) day-by-day test data.

Day	B40 (NmLCH_4_/Bott)	B40Y (NmLCH_4_/Bott)	B15Y (NmLCH_4_/Bott)	Cellulose, (NmLCH_4_/Bott)
1	69.68	70.63	52.95	10.22
2	347.00	361.25	340.92	239.50
3	601.88	609.45	660.87	542.92
4	686.07	687.85	747.11	673.55
5	724.89	741.37	799.03	747.39
6	744.59	761.42	820.39	786.53
7	755.25	778.25	832.29	798.51
8	765.84	784.55	838.88	803.03
9	772.87	793.32	844.13	805.57
10	780.03	797.97	848.36	807.62
11	791.29	807.95	855.32	810.12
12	797.85	815.66	862.18	812.37
13	804.10	821.49	870.09	816.28
14	806.68	825.04	872.25	817.50
15	809.21	827.71	874.99	819.52
16	810.40	829.31	876.45	821.64
17	811.65	830.56	877.78	822.96
18	814.20	832.85	880.50	825.70
19	817.28	835.21	880.92	827.15
20	818.28	836.08	881.32	827.15
21	818.77	836.91	881.64	827.17
22	819.63	838.04	882.00	827.54
23	820.93	839.27	882.33	827.83
24	821.98	840.53	882.69	828.13
25	822.84	841.48	882.93	828.23
26	823.73	842.44	883.16	828.34
27	824.72	843.42	883.40	828.44
28	825.64	844.85	883.58	828.54
29	826.27	846.28	883.58	828.65
30	826.27	847.40	883.58	828.86

## Data Availability

The authors declare that the data supporting the indings of this study are available within the paper.

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
