# Peer review of "Wheat Bran and Saccharomyces Cerevisiae Biomass’ Effect on Aerobic and Anaerobic Degradation Efficiency of Paper Composite"

_microorganisms, 2024, doi:10.3390/microorganisms12102018_

Round 1

Reviewer 1 Report

Comments and Suggestions for Authors

This manuscript deals with the study of the biodegradation efficiency of paper composites of different compositions under aerobic and anaerobic conditions.

After careful reading, I judge that there are some serious issues that need to be revised:

 Abstract

The Abstract should be rewritten after the manuscript has been revised.

 Introduction

1. The Introduction generally describes the problem.

2. “However, the validity of the aerobic test is compromised due to the insufficient degradation of the reference material, indicating the need for further testing under improved conditions to ensure accurate and reliable results” - The authors should cite literature data on the extent of cellulose degradation, otherwise this statement should be rephrased.

 Materials and methods

1. On page 2 there is a table without numbering. In addition, another sample (B15) is listed, but the results of experiments with this sample are not presented.  The authors should add these data.

In the same table, the species name should be given in italics.

2. Chapter 2.4. Authors should report the pH at the start of the test.

3. Table 2. “Anerobic” should be replaced by” Anaerobic”.

 Results and Discussion

1. Tables 3 and 4 do not include error data, but the experiments were performed in three replicates.

2. Page 5. (“see Chapter 2.4”) - Perhaps the authors had Section 2.5 in mind?

3. Page 6. (“According to standard practices, such low biodegradation rates suggest that the test environment, inoculum, or equipment may not have been optimal for supporting aerobic microbial activity. As a result, the aerobic biodegradation efficiency of other materials tested in this environment may also be un[1]derestimated”) - References should be cited.

 Conclusions

This section should be reduced and merged with the Future research section (taking into account the comment on Future research).

 Future research

The text from this section should be reduced to 1-2 sentences and moved to the Conclusions section.

Comments on the Quality of English Language

Minor editing of English language required.

Author Response

(Reviewer 1)

This manuscript deals with the study of the biodegradation efficiency of paper composites of different compositions under aerobic and anaerobic conditions.

ANSWER

Dear Reviewer,

First, thank you so much for your time and valuable insights. Please find below the explanations and corrections made in response to your comments.

--------------------------------------

After careful reading, I judge that there are some serious issues that need to be revised:

Abstract

The Abstract should be rewritten after the manuscript has been revised.

ANSWER After reviewing the manuscript thoroughly, no major changes were made that would affect the research results and the information provided in the abstract.

--------------------------------------

Introduction

  1. The Introduction generally describes the problem.

ANSWER Thank you.

  1. “However, the validity of the aerobic test is compromised due to the insufficient degradation of the reference material, indicating the need for further testing under improved conditions to ensure accurate and reliable results” - The authors should cite literature data on the extent of cellulose degradation, otherwise this statement should be rephrased.

ANSWER Corrections are made and added to 3.1 accordingly: Aerobic biodegradation test validity: the validity of the aerobic test is questionable due to the exceptionally low biodegradation efficiency of the reference material (cel-lulose), which degraded by only 0.71% over 30 days. According to ISO 14855-2 [27], the test is considered valid only if the degree of biodegradation of the reference material exceeds 70% after 45 days. Given that the biodegradation rate falls significantly below this threshold, the low biodegradation rates suggest that the test environment, inoculum, or equipment may not have been optimal for supporting aerobic microbial activity. Consequently, the aerobic biodegradation efficiency of other materials tested in this environment may also be underestimated.

--------------------------------------

Materials and methods

  1. On page 2 there is a table without numbering. In addition, another sample (B15) is listed, but the results of experiments with this sample are not presented. The authors should add these data.

In the same table, the species name should be given in italics.

ANSWER Corrected, data added. Thank you.

--------------------------------------

  1. Chapter 2.4. Authors should report the pH at the start of the test.

ANSWER pH value added. Thank you.

--------------------------------------

  1. Table 2. “Anerobic” should be replaced by” Anaerobic”.

ANSWER Corrected, thank you.

--------------------------------------

Results and Discussion

  1. Tables 3 and 4 do not include error data, but the experiments were performed in three replicates.

ANSWER Information, allowing for a maximum error of 1%, is added. Information, allowing for a maximum error of 1%, is added. We have performed these experiments by making very precise measurements, allowing us to minimize variability and ensure data accuracy. For example, the following table shows sample weights with high precision:

sample g.

Ref

1.0034

Ref

1.0031

Ref

1.0018

12

NBSK B40

1.0063

12

NBSK B40

1.0084

12

NBSK B40

1.0000

13

NBSK B40 Y

1.0013

13

NBSK B40 Y

1.0027

13

NBSK B40 Y

1.0019

  1. Page 5. (“see Chapter 2.4”) - Perhaps the authors had Section 2.5 in mind?

ANSWER Yes, thank you.

--------------------------------------

  1. Page 6. (“According to standard practices, such low biodegradation rates suggest that the test environment, inoculum, or equipment may not have been optimal for supporting aerobic microbial activity. As a result, the aerobic biodegradation efficiency of other materials tested in this environment may also be un[1]derestimated”) - References should be cited.

ANSWER ISO standard and additional reference are added, thank you.

--------------------------------------

Conclusions

This section should be reduced and merged with the Future research section (taking into account the comment on Future research).

Future research

The text from this section should be reduced to 1-2 sentences and moved to the Conclusions section.

ANSWER Corrections are made:

The comparison of aerobic and anaerobic degradation: The performed tests reveal dis-tinct differences in efficiency and outcomes. Under aerobic conditions, the biodegrada-tion efficiency is significantly lower compared to anaerobic biodegradation, with values ranging from 0.71% to 6.34%. B40Y exhibits the highest aerobic efficiency at 6.34%, followed by B40 (4.07%) and B15Y (2.24%). Cellulose shows the lowest efficiency at 0.71%.

In contrast, anaerobic biodegradation shows much higher efficiencies, with bio-degradation percentages reaching up to 96.62% for B15Y, while B40 and B40Y demon-strate slightly lower values of 88.50% and 88.14%, respectively. However, these values are still significantly higher compared to the control (cellulose, 82.32%).

This stark difference can be attributed to the metabolic capabilities of anaerobic microbes, which efficiently convert organic matter into biogas, particularly methane (CH₄).

Aerobic biodegradation (46°C, Mesophiles and Thermophiles): Higher concentrations of wheat bran lead to higher biodegradation efficiencies due to the rich non-starch poly-saccharides and essential nutrients present in wheat bran, which support and stimulate both mesophilic and thermophilic microbial activity.

Anaerobic biodegradation (55°C, Thermophiles): Thermophiles at 55°C have fewer metabolic pathways available for degrading organic matter without oxygen. The breakdown of organic matter is slower, and the presence of lignin in wheat bran further complicates the process due to its resistance to microbial degradation.

The addition of yeast: even in an inactive form yeast additive provides additional nutrients that might stimulate microbial growth and activity, enhancing biodegradation efficiency under aerobic conditions and having unsignificant positive impact in anaerobic conditions.

Future research: due to the observed exceptionally low aerobic biodegradation efficiency, future research will focus on optimizing tests conditions, testing materials at a temperature range of 50-60°C, using compost with a 40%-60% moisture content as the inoculum. Advanced testing equipment with automatic oxygen supply will be utilized.

--------------------------------------

Thank you again for your time and insights!

Have a great autumn,

Zita Markeviciute

Reviewer 2 Report

Comments and Suggestions for Authors

The manuscript "Wheat bran and Saccharomyces cerevisiae biomass effect on aerobic and anaerobic degradation efficiency of papeer composite" shows some interesting results regarding the biodegradation of paper composite. However, some improvements must be undertaken before considering the manuscript ready for publication.

I find the experimental design short. I would like to see a sequential increase in bran content, not only 15 and 40, so a more accurate estimation of the best concentration would be done.

Authors discuss the presence of lignin affecting the biodegradation. It would be interesting to measure the lignin concentration and to stablish its relation with biodegradation rates.

Error bars should be added to the figure and error should be shown in tables. Also, time series biodegradation would also be interesting to check in a figure, not only the final measure. Check the biodegradation curve could give an estimation about the biodegradation speed and if it continues or gets a plateau at the end of the experiment.

There are many paragraphs without or very few references along the discussion and conclussions.

Control samples should be included in the results.

If the main aim of the research is to identify the optimal conditions to enhance the biodegradation, it would be interesting to identify which microbes are responsible of this degradation with the different conditions measured, so authors could modify specifically the medium provided for the biodegradation.

Authors take a biodegradation rate of ~6% as very significant with aerobic conditions, but I don't think that this is a result that could be considered so successful, considering the high rates found with anaerobic biodegradation. Authors should discuss better the high difference between the two biodegradation types. 

Author Response

TO: Reviewer 2

The manuscript "Wheat bran and Saccharomyces cerevisiae biomass effect on aerobic and anaerobic degradation efficiency of paper composite" shows some interesting results regarding the biodegradation of paper composite. However, some improvements must be undertaken before considering the manuscript ready for publication.

ANSWER

Dear Reviewer,

First, thank you so much for your time and valuable insights. Please find below the explanations and corrections made in response to your comments.

--------------------------------------

  1. I find the experimental design short. I would like to see a sequential increase in bran content, not only 15 and 40, so a more accurate estimation of the best concentration would be done.

ANSWER

This study represents ongoing research aimed at creating biodegradable packaging material using wood pulp, agricultural by-products (to conserve raw materials), and a natural bio-coating with Saccharomyces cerevisiae biomass.

For the initial sample trials during the sheet formation and production stages, we selected 15% and 40% concentrations of fillers to evaluate their impact on physical-mechanical properties first https://link.springer.com/article/10.1007/s43621-024-00257-8

Two different fillers were tested: wheat bran, which is presented in this study, and wheat grain residues, which exhibited lower physical and mechanical properties in earlier stages of our research. We also incorporated 10 wt% Saccharomyces cerevisiae as a bio-additive. In total, more than ten different formulations were tested.

Three materials that demonstrated the best hydrophobic properties were selected for further biodegradation testing. This was done to observe the general trends related to the impact of additives and fillers. The results from these trials will guide the next phase of research, which will involve optimizing material compositions and evaluating their physical-mechanical properties as well as biodegradation efficiency under different conditions.

--------------------------------------

  1. Authors discuss the presence of lignin affecting the biodegradation. It would be interesting to measure the lignin concentration and to stablish its relation with biodegradation rates.

ANSWER

The lignin content of bran  was indicated in our previous study https://www.mdpi.com/2071-1050/14/12/7393 Given the lignin's well-known resistance to microbial action, it is reasonable to assume that the lignin concentration in the tested paper composites correlates negatively with biodegradation efficiency. Higher bran content increases lignin levels, which can lead to slower degradation and reduced methane production, as seen in our data.

The trend of lower biodegradability with increasing lignin content is well known from already established scientific findings on lignin's role in inhibiting microbial decomposition.

Also – additional reference nr. [55] is added.   

--------------------------------------

3.1 Error bars should be added to the figure and error should be shown in tables.

ANSWER

Information, allowing for a maximum error of 1%, is added. Information, allowing for a maximum error of 1%, is added. We have performed these experiments by making very precise measurements, allowing us to minimize variability and ensure data accuracy. For example, the following table shows sample weights with high precision:

sample g.

Ref

1.0034

Ref

1.0031

Ref

1.0018

12

NBSK B40

1.0063

12

NBSK B40

1.0084

12

NBSK B40

1.0000

13

NBSK B40 Y

1.0013

13

NBSK B40 Y

1.0027

13

NBSK B40 Y

1.0019

3.2 Also, time series biodegradation would also be interesting to check in a figure, not only the final measure. Check the biodegradation curve could give an estimation about the biodegradation speed and if it continues or gets a plateau at the end of the experiment.

ANSWER

Day-by-day net cumulative gas production table (Table 6. Mean net cumulative gas production (NmL CH4) day by day test data) is added. Thank you for this insight.

--------------------------------------

  1. There are many paragraphs without or very few references along the discussion and conclusions.

ANSWER

Additional references are added [49], [55].  

--------------------------------------

  1. Control samples should be included in the results.

ANSWER

Cellulose was used as a control sample. This information is provided in 2.4 paragraph - Microcrystalline cellulose was used as a reference material. 

--------------------------------------

6.1 If the main aim of the research is to identify the optimal conditions to enhance the biodegradation, it would be interesting to identify which microbes are responsible of this degradation with the different conditions measured, so authors could modify specifically the medium provided for the biodegradation.

ANSWER

Unfortunately at the labs where biodegradation tests were performed there was no equipment to indicate microbiological diversity.

6.2 Authors take a biodegradation rate of ~6% as very significant with aerobic conditions, but I don't think that this is a result that could be considered so successful, considering the high rates found with anaerobic biodegradation.

Thank you for this comment. To clarify what was meant corrections are added: The comparison of aerobic and anaerobic degradation: performed tests reveals distinct differences in efficiency and outcomes. Under aerobic conditions, the biodegradation efficiency, compared to anaerobic biodegradation, is significantly lower, with values ranging from 0.71% to 6.34%. B40Y has the highest efficiency at 6.34%, followed by B40 (4.07%), and B15Y (2.24%). Cellulose has the lowest efficiency at 0.71%.

In a contrast, anaerobic biodegradation shows much higher efficiencies, with biodegradation percentages reaching up to 96.62%. B15Y has the highest biodegradation efficiency at 96.62%, significantly higher than cellulose (82.32%). Both B40 and B40Y have similar efficiencies (88.50% and 88.14%, respectively) but are still higher than cellulose.

6.3 Authors should discuss better the high difference between the two biodegradation types.

ANSWER

The comparison of aerobic and anaerobic biodegradation types are already described in many scientific literature, so we decided not to provide this information.   

--------------------------------------

Thank you again for your time and insights!

Have a great autumn,

Zita Markeviciute

Round 2

Reviewer 2 Report

Comments and Suggestions for Authors

Authors have addressed almost all my comments and improved the manuscript accordingly. Perhaps, authors could add in the methodology section a small summary of what was done in the previous work. A short two lines mention would save readers to have to check the whole previous work and have a better understanding of what is the starting point of this manuscript.